# Effects of Yoga for Coping with Premenstrual Symptoms in Taiwan—A Cluster Randomized Study

**DOI:** 10.3390/healthcare11081193

**Published:** 2023-04-21

**Authors:** Hsing-Chi Chang, Yi-Chuan Cheng, Chi-Hsuan Yang, Ya-Ling Tzeng, Chung-Hey Chen

**Affiliations:** 1Department of Nursing, National Taichung University of Science and Technology, Taichung 404336, Taiwan; celiaproactive4142@gm.nutc.edu.tw (H.-C.C.);; 2Department of Nursing, Asia University, Taichung 41354, Taiwan; 3School of Nursing, College of Healthcare, China Medical University, Taichung 404328, Taiwan; 4Department of Nursing, Hungkuang University, Taichung 433304, Taiwan; 5Department of Nursing & Institute of Allied Health Sciences, National Cheng Kung University, Tainan 70101, Taiwan

**Keywords:** premenstrual symptoms, depressive symptoms, physical symptoms, anger, yoga

## Abstract

Home-based yoga practice has not been approved as a method for alleviating premenstrual symptoms in Taiwan. This study was a cluster randomized trial. A total of 128 women self-reporting at least one premenstrual symptom were enrolled in the study, of which there were 65 participants in the experimental group and 63 participants in the control group. Women in the yoga group were provided with a yoga DVD program (30 min) to practice for three menstrual months, at least three times a week. All participants were given the Daily Record of Severity of Problems (DRSP) form to measure premenstrual symptoms. After the yoga exercise intervention, the yoga group had statistically significantly fewer and/or less severe premenstrual depressive symptoms, physical symptoms, and anger/irritability. Other disturbances and the impairment of daily routine, hobbies/social activities, and relationships also occurred significantly less often in the yoga group. The study found that yoga is useful to relieve premenstrual symptoms. Moreover, home-based yoga practice is more pertinent in the pandemic era. The strengths and drawbacks of the study are discussed and further study is recommended.

## 1. Introduction

Women of reproductive age experience premenstrual symptoms, including physical discomfort, mood swings and depressed symptoms, during the luteal phase of menstruation [1]. The reason for premenstrual symptoms remains unclear and may be multifaceted. It is suggested that premenstrual symptoms are caused by underlying neurobiological vulnerability to normal fluctuations in circulating gonadal hormones during the menstrual cycle [2,3]. Previous studies indicate that the prevalence of premenstrual symptoms ranges from 48 to 90% worldwide and that about 20% of women experience premenstrual symptoms severe enough to affect their quality of life [4,5,6]. Premenstrual symptoms have numerous influences, including physical and emotional symptoms, thus impacting everyday life, regular activities, and interpersonal relationships. To reiterate, premenstrual symptoms, mild or severe, cause varying degrees of disturbance [7,8]. Surveys on the disturbance women encounter disclosed that women with premenstrual symptoms engage in negative behaviors, including shouting, impatience, throwing things, negative communication, or even physically abusing children [9,10]; it may cause women to be less efficient in their learning or work, with higher rates of absenteeism and limited productivity [10,11]. Furthermore, studies revealed that premenstrual symptoms could lead to less interest in activities and negative interpersonal relationships [12,13].

The treatment of premenstrual symptoms involves medical and non-medical therapy. Compared with medical therapy, which may have side effects, non-medical therapy is more acceptable. Non-medical therapies generally include alimentary therapeutics [14,15] as well as complementary and alternative medicine such as rest, heat compress, traditional Chinese medicine, lifestyle changes, meditation, regular exercise, etc. [16,17,18,19], which are commonly used. Many recent studies have discussed the effect of exercise on premenstrual symptoms [20,21,22], mainly because it is not only convenient, non-invasive, and has low side effects but also maintains overall health. Exercise is still worthwhile as an essential strategy for alleviating pre-menstrual symptoms, even though its actual effectiveness in improving pre-menstrual symptoms awaits confirmation [23].

Studies have shown that regular exercise can minimize fluctuations in the concentration of estrogen and other steroid hormones, reducing their physical and emotional effects [24]. Specifically, regular moderate exercise can reduce cortisol [25], increase oxygenation [16,26], and raise endorphins [27], consequently ensuring women’s physical and mental comfort. The American College of Obstetricians and Gynecologists and National Health Service recommends that performing stretching and breathing exercises [28,29], such as yoga and pilates, could help women sleep better and reduced stress levels during the late luteal period. Yoga is regarded as a psychophysical exercise [30,31]. The discipline is helpful for concentration, consciousness, and physical flexibility [32,33], which brings about physically and mental integration.

Yoga denotes union and integration and improves biopsychosocial and spiritual wellness. In other words, yoga is a form of mind–body health that combines physical activity with inner-directed mindfulness to achieve self-consciousness through breathing and meditation [34]. Contemporarily, the practice of yoga includes three essential elements: breath control (pranayama), postures (asana), and meditation (dhyana). This triad of characteristics makes yoga more tangible and accessible for people to understand and practice [35,36].

Breath control (pranayama) should be integrated into the practice of postures (asanas) and meditation. This enables individuals to observe changes in the body through various postures and mindful breathing, and thus to adjust self-consciousness accordingly to achieve physical and mental balance [37,38]. Studies have shown that breathing techniques can regulate the function of the autonomic nervous system, inhibit the sympathetic nervous system activity, reduce the heart rate and blood pressure, diminish energy consumption, and increase theta and delta brain waves [39,40,41]. These effects can promote relaxation of the mind and body and may help alleviate feelings of tension, anxiety, or depression. Postures (asana) in yoga are the imitation of original forms in nature and animal movements, which are then modified to create various movements to enhance physical function [42]. These movement patterns resemble isometric contractions, and the practice mode is based on an intermittent pattern (static and dynamic intermittently) [43]. The mechanism by which postures (asanas) affect the body involves the positioning of limbs and the contraction of muscles, which stimulate the pressure receptors located beneath the skin. This, in turn, leads to an increase in the activity of the vagus nerve, reducing the production of cortisol [44]. These physiological and biochemical changes have a positive impact on pain, depression, and immune function. Meditation (dhyana) is a continuous state of mindfulness, with the aim of achieving inner peace and tranquility. In empirical research on yoga, meditation is also referred to as relaxation training or relaxation exercises. Through consciously controlling autonomic psycho-physiological activity, lowering arousal levels, relaxation training can help to relax the mind [45,46]. Meditation stimulates and activates alpha brain waves and is beneficial in alleviating emotional tension. Additionally, it can increase the activity of theta waves and promote the secretion of melatonin, which not only aids in falling asleep but also improves the quality of sleep.

To summarize, knowledge on how to alleviate the complex effects of premenstrual symptoms is crucial and beneficial for women’s overall health and well-being. Any approach that can provide relief for these symptoms can help women achieve biopsychosocial wellness. In Taiwan, yoga exercises personally instructed by a certified and experienced yoga instructor have been shown to alleviate premenstrual symptoms among women [47]. However, no study has yet been conducted to investigate whether a home-based yoga practice, using a guided yoga DVD program, can effectively alleviate these symptoms. Therefore, the aim of this study is to examine the effectiveness of home-based yoga practice for alleviating premenstrual symptoms among women in the community.

## 2. Materials and Methods

### 2.1. Study Design and Sample

This study was a cluster randomized trial with two groups. The intervention was yoga exercise three times (30 min per section) a week for 3 months under instruction using compact disc multimedia. After the institutional review board approved the study protocol, participants were recruited from the Betun District and Nantun District in Taichung. The study participants were recruited using a poster, which read “Do you feel emotional ups and downs, overwhelmed or uncomfortable frequently before menses?” Considering age is one of the impact factors on women’s physiological states, including menstruation, and endocrine stability, only women aged 20–40 were recruited.

A convenience sample of women who had self-reported at least one of premenstrual symptoms, such as headache, breast swelling, mood swings, fatigue, etc., was recruited. The inclusion criteria were women who: (1) had regular menses a with menstrual cycle length of 21–35 days and (2) were able to communicate in Mandarin and fill in the questionnaire. The exclusion criteria were women who: (1) were using contraception, e.g., oral contraceptives or an intra-uterine device (IUD); (2) were receiving infertility treatment or hormone therapy; (3) had experienced menopause, such as having had a hysterectomy or ovariectomy; (4) had given birth in the latest year; and/or (5) had been diagnosed as having endocrine disturbances.

Women who met the inclusion criteria were given an explanation of the study and needed to be willing to complete a daily diary for three menstrual cycles. In order to avoid women in the two treatment groups inadvertently sharing information on the intervention (yoga exercise), a cluster design employing two geographically distinct districts to recruit for the experimental and control groups was utilized.

According to the original Daily Record of Severity of Problems (DRSP) developer [48] as well as several research studies using DRSP for measures, the dropout rate of study subjects is typically about 25% [48,49,50]. Considering the sample/sampling errors and confidence level of 0.95, the estimation of the study sample size was 80 women in each group, i.e., 160 women were included in the study. The Institutional Review Board, National Cheng Kung University Hospital, approved the study (B-ER-104-042). Written and signed informed consent was obtained from each participant.

### 2.2. Measures

The Daily Record of Severity of Problems (DRSP) was used to capture premenstrual symptoms. DRSP contained two parts: symptoms related to menstruation (21 items) and their impairment in daily life (3 items). The symptoms included depression, hopelessness, feelings of worthlessness, increased sleeping, trouble sleeping, feeling overwhelmed, breast tenderness, breast swelling, headache, joint or muscle pain, anger, conflicts or problems with people, anxiety, mood swings, sensitivity to rejection, decreased interest, difficulty concentrating, fatigue, increased appetite, craving specific foods, and feeling out of control. The impairments caused by the symptoms pertained to the daily routine, hobbies/social activities, and interpersonal relationships. All items were rated from 1 (“not at all”) to 6 (“extreme”) [48]. The higher the score, the more severe the premenstrual symptoms. The internal consistency (Cronbach alpha) of DRSP was 0.96, and its concurrent validity with the Hamilton Depression Rating Scale score was highly correlated (r = 0.75) [51]. The demographic data were measured too.

### 2.3. Intervention

A yoga DVD program was given to the yoga group, which was composed by a certified yoga instructor who had received yoga physiotherapist training. The Yoga DVD program lasted 30 min in total, which included the following elements [31,52,53]:◆ Five minutes for warming up from head to belly.◆ Twenty minutes of asana practice, including Surya Namaskar (sun salutation), Supta Baddha Konasana (reclining bound angle pose), Supta Padangusthasana (reclining hand-to-big-toe pose), Upavistha Konasana (seated wide-angle pose), and Psrsva Virasana (side twisted hero pose).◆ Five minutes of Ardha Padmasana (half lotus pose).

Participants in the yoga group were requested to use the yoga DVD program to practice at least three times a week, for three menstrual cycles. A question about whether participants had engaged in yoga exercise today was added to the questionnaire to remind the participants to practice. A research assistant called the participants once a week to ensure that they performed the regimen as prescribed/told. When called, the participants were also encouraged to ask questions related to the yoga practice and keeping the log.

### 2.4. Data Collection and Analysis

After signing the informed consent, the participants were asked to complete the demographic data form. Then, they were given the Daily Record of Severity of Problems (DRSP) form to fill out. Each participant was requested to complete the DRSP form according to their individual experiences before going to bed each day. The DRSP form consisted of items structured with scales and dates, which enabled participants to simply check off the corresponding cells. At the conclusion of each menstrual cycle, participants returned the completed questionnaires using a prepaid envelope. This process was repeated at the end of the second and third menstrual cycles. Data collection occurred between March and December 2015.

Data from the late corpus luteum phase (5 days before menstruation) were used in this paper. The scores for each individual symptom were calculated as the sum over the 5-day period. The statistical analysis was performed using SPSS software (Version 21.0, IBM Corp., Armonk, NY, USA) [54]. An independent *t*-test was employed to determine the effects of yoga exercise on premenstrual symptoms. The accepted level of statistical significance for all analyses was set at *p* < 0.05.

## 3. Results

A total of 180 women were enrolled in this study. As illustrated in Figure 1, 15 women in the yoga group were excluded, including 6 who became pregnant, 5 who traveled abroad, and 4 who did not complete the questionnaires. In the control group, 17 women were excluded, including 7 who became pregnant, 7 who were not willing to participate, and 3 who did not complete the questionnaires. Ultimately, there were 65 participants (age range, 32–36 years; mean age: 31.60 years) in the experimental group and 63 participants (age range, 30–36 years; mean age: 30.35 years) in the control group who completed the study, a total of 128 women (Figure 1).

Demographic data for the two groups are presented, and no significant differences were observed between the groups in terms of demographic characteristics, including education level, occupation, shift work, marital status, physical exercise habits, age at first menstruation, duration of menstruation, amount of menstrual bleeding, premenstrual pain, and menstrual pain (Table 1). There were no statistically significant differences in the baseline Daily Record of Severity of Problems (DRSP) scores between the two groups.

Following the yoga exercise intervention, the yoga group exhibited statistically significant reductions in premenstrual symptoms related to depression, physical symptoms, and anger/irritability, as compared to the control group. Additionally, the yoga group demonstrated significantly lower levels of other emotional disturbances, including anxiety, mood swings, sensitivity to rejection, reduced interest, difficulty concentrating, lethargy, increased appetite, specific food cravings, and feeling out of control, as compared to the control group (*p* < 0.05). Moreover, impairments caused by premenstrual symptoms, such as difficulties with work or daily routine, hobbies or social activities, and relationships with others, were found to be significantly lower in the yoga group as compared to the control group (Table 2).

## 4. Discussion

Some authors posit that the benefits of yoga are almost the same as or even better than other exercises in terms of its effect on physical and mental health [44,55]. Specifically, studies have shown that regular exercise can reduce premenstrual physical and psychological symptoms. The result is similar to the findings of past randomized trials with yoga [26,50,56,57]. In our research, we found that yoga exercises for three months could effectively alleviate premenstrual symptoms, including depressive symptoms [58,59,60], physical symptoms [47,58,60], and anger/irritability [60,61]. Other disturbances, including feeling anxious, sensitive to rejection, and out of control, and experiencing mood swings, lowered interest, difficulty concentrating, lethargy, increased appetite, and cravings for specific foods, were significantly decreased after three months of yoga intervention. These findings are partially supported by other research [41,60,62], in which it was found that symptoms including anxiety [60,62], fatigue [60], reduced interest, difficulty concentrating [32,41], and changes in appetite [60] were improved in the yoga group.

As a bio-psychosocial being, menstruation is an essential part of a woman’s physiology. Inevitably, some women suffer from premenstrual symptoms. Yoga exercises are known to affect the body through physical and mental integration [30,31,44,55]. Regular yoga practice can help women achieve physical and mental relaxation and freedom from disturbances [57]. Our study found that after three months of yoga practice, participants successfully reduced the impact of premenstrual symptoms on their work, daily routine, hobbies, social activities, and relationships. These findings are consistent with previous studies [47,60,63,64,65] indicating that yoga can improve physical and mental wellness and reduce the impact of symptoms on women’s daily life.

Studies show that aerobic exercises such as walking, running, and swimming are effective in reducing premenstrual symptoms [66,67]. Moreover, Pilates [68] and Baduanjin, a traditional Chinese medical exercise [50], were found to be able to mitigate premenstrual symptoms. In addition, one study was conducted to compare the effect of acupressure and yoga on premenstrual symptoms, revealing that yoga was more effective in decreasing the severity of premenstrual symptoms [60]. All in all, in light of the attribute physical and mental integration, this study’s results echo these research findings.

Methodologically, the daily record of severity of problems (DRSP) daily diary was used to capture women’s premenstrual symptoms. Keeping a daily log can be viewed as a self-awareness checklist for women who may not yet realize that their symptoms are related to the menstrual cycle, which is physiologically normal. This approach can be beneficial for women to better understand their bodies, both in physiological and emotional states. The authors’ intention is to help women cope effectively with premenstrual symptoms and avoid being diagnosed with premenstrual syndrome (PMS). The DRSP is presently acknowledged as the most effective tool for assessing and monitoring premenstrual symptoms. According to research, the DRSP is dependable and valid in evaluating the severity and fluctuations of premenstrual symptoms [48,49,69]. Moreover, the daily tracking of symptoms with the use of DRSP can measure physiological, psychological, and behavioral changes during different phases of the menstrual cycle in women [48,51]. The findings of this study demonstrate that yoga can alleviate premenstrual symptoms in the luteal phase of the menstrual cycle. However, the authors do not show the data changes during other phases of the menstrual cycle; hence, the effect of yoga across each phase cannot be outlined.

Most research studies related to yoga exercises lasted around 3 months [44,47,60]. However, whether or not the length of practicing yoga is enough for interpreting research findings is still contested. Practicing yoga benefits women not only by reducing premenstrual symptoms but also by promoting women’s physical and mental integration, which improves their quality of life. Therefore, yoga exercises are worth practicing for a longer time, which would be more statistically significant and practically significant. However, due to the lack of further follow-up after the intervention, the long-term benefits of the exercise intervention in this study cannot be confirmed.

Good compliance with the regimen corresponded to the exercise effects [70,71]. Considering that it is convenient for the participants to choose the time and place feasible to exercise, the authors designed a yoga DVD program that guided the participants to practice at home. Moreover, at the end of the DRSP form, one question about whether exercise was carried out today was added to remind the participants to continue the yoga program. Furthermore, each participant received a weekly call from a research assistant enquiring about both the yoga exercise performance and whether they had filled out the DRSP form, as well as answering any enquiries regarding these.

What is mentioned above contributed to the compliance of the participants with the yoga practice, in turn influencing the yoga effects. However, the authors failed to record the compliance of yoga practice. As one study noted [72], home training by video and self-reporting could make compliance problematic. Consequently, the confounding role of the compliance of yoga practice on decreasing premenstrual symptoms warrants further study. Moreover, the daily record, written reminder, and weekly calling might also bring about suggestive effects. The interaction of those confounding factors should be clarified, and then the study replication and applicability could be improved as one study [73] recommended.

One of the main strengths of this study was the meticulous use of the Daily Record of Severity of Problems (DRSP). By requiring participants to record their symptoms on a daily basis, the study effectively minimized the potential for recall bias, which can often be a significant limitation in studies of this nature. The DRSP questionnaire proved to be an effective tool for capturing a comprehensive picture of each participant’s symptoms throughout the study period. In addition to the use of the DRSP questionnaire, the provision of a yoga DVD for convenient practice was also a key strength of the study, especially in light of the COVID-19 pandemic, as it allowed for yoga practice without face-to-face interaction. Moving forward, it may be worth exploring the use of smartphone apps for collecting DRSP data, as this could help to further enhance the accuracy and completeness of the data collected.

The study has several limitations. Firstly, the generalizability of the results is limited to women within the community where the study was conducted. Secondly, the lack of physiological data to verify the effect of yoga exercise is another limitation. Including such data, as well as information on the extent of medication use by participants, would have led to a better interpretation of the results. Thirdly, although participants were advised not to engage in other forms of exercise during the three menstrual months of the study, there was no item in the questionnaire to monitor other exercises. Another potential limitation of the study was the use of pen and paper to collect DRSP records, which could have resulted in missing or incomplete data. Exploring the use of smartphone apps to collect DRSP data could be beneficial to address this limitation [74]. This approach could reduce the rate of missing data and increase the percentage of completers. Additionally, digital records could be easily stored, organized, and analyzed, leading to more efficient data management.

## 5. Conclusions

Yoga has been shown to provide numerous physical and mental health benefits. This study confirms that yoga can effectively reduce premenstrual symptoms, including depressive symptoms, physical symptoms, and anger/irritability, as well as their impact on daily routines, hobbies/social activities, and relationships. Yoga is a safe, simple, and cost-effective approach that can be practiced alone anytime and anywhere for health purposes. Women are recommended to practice yoga as a way to cope with premenstrual symptoms. Furthermore, the thoughts and reflections of women during the yoga intervention and while responding to the questionnaire are worth investigating. A qualitative research approach could be utilized to explore the manifestation of women’s responses during yoga practice.

## Figures and Tables

**Figure 1 healthcare-11-01193-f001:**
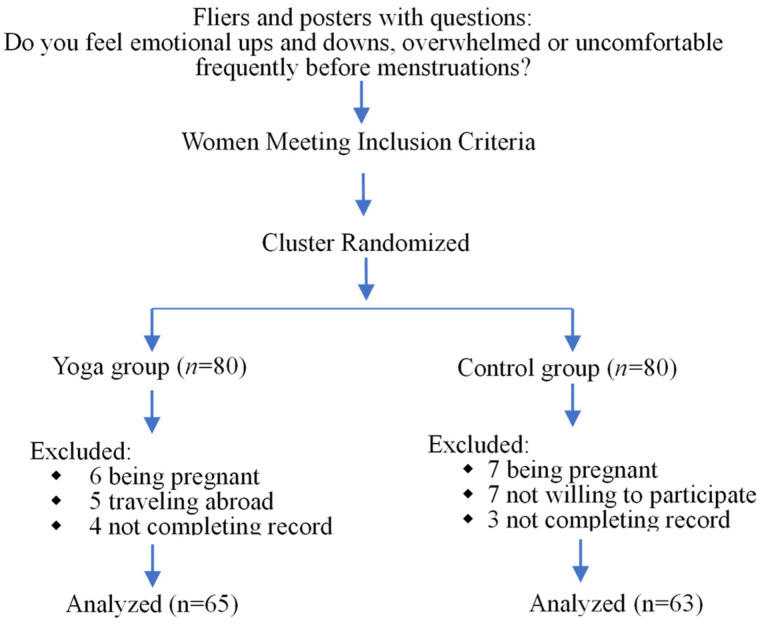
Flow diagram of participant enrollment.

**Table 1 healthcare-11-01193-t001:** Characteristics of the participants in both groups.

Characteristics	Yoga Group (*n* = 65)		Control Group (*n* = 63)		*t/* *χ^2^*	*p*
	M ± SD/*n* (%)		M ± SD/*n* (%)			
Physical exercise (exercise habit)					0.76	0.38
Yes	27	(41.54)	31	(49.21)		
No	38	(58.46)	32	(50.79)		
Days of bleeding during menses	5.55	±1.24	5.62	±1.17	−0.31	0.76
Menstruation amount					1.48	0.48
Little	0	(0.00)	1	(1.59)		
Moderate	59	(90.77)	54	(85.71)		
Heavy	6	(9.23)	8	(12.70)		
Pre-menstrual pain					0.47	0.49
Yes	45	(69.23)	40	(63.49)		
No	20	(30.77)	23	(36.51)		
Menstrual pain					2.70	0.44
No	14	(21.54)	15	(23.81)		
Rarely	15	(23.08)	14	(22.22)		
Sometimes	21	(32.31)	26	(41.27)		
Regularly	15	(23.08)	8	(12.70)		

Note. Values are expressed as mean ± standard deviation or number (percentage).

**Table 2 healthcare-11-01193-t002:** Comparison of premenstrual symptoms of the participants in both groups.

Variable	Baseline				*p*	After Intervention				*p*
	Yoga (*n* = 65)		Control (*n* = 63)			Yoga (*n* = 65)		Control (*n* = 63)		
	Mean	SD	Mean	SD		Mean	SD	Mean	SD	
Depressive symptoms	52.90	22.48	46.90	17.56	0.23	39.49	7.90	49.17	14.56	0.00 ***
Depressed/sad/blue	9.20	4.68	7.55	4.37	0.13	6.49	2.01	7.11	3.90	0.26
Hopeless	6.38	3.23	6.00	2.49	0.59	5.48	1.00	5.94	2.24	0.14
Worthless/guilty	6.73	3.69	6.10	2.30	0.41	5.52	1.03	6.11	2.66	0.11
Slept more	11.35	5.97	10.77	5.62	0.68	8.25	3.29	11.70	4.51	0.00 ***
Trouble sleeping	10.13	6.03	7.94	4.43	0.09	6.97	2.77	8.16	4.31	0.07
Overwhelmed, can’t cope	9.23	4.81	8.48	3.55	0.47	6.78	2.31	10.16	4.00	0.00 ***
Physical symptoms	43.83	20.98	39.97	14.65	0.39	30.37	8.62	40.84	11.32	0.00 ***
Breast tenderness	11.60	6.57	10.97	4.51	0.65	7.82	3.05	11.60	4.77	0.00 ***
Breast swelling/“bloated”	12.15	6.90	11.87	4.30	0.84	8.14	3.05	12.40	4.52	0.00 ***
Headache	9.70	5.92	8.71	4.27	0.43	6.55	1.86	8.43	2.73	0.00 ***
Joint or muscle pain	10.38	5.77	8.45	4.49	0.13	7.86	2.87	8.41	3.80	0.36
Anger/irritability	15.95	6.98	15.23	4.72	0.62	12.52	3.01	16.22	5.44	0.00 ***
Angry, irritable	8.70	4.29	8.23	3.01	0.60	6.68	2.31	8.98	3.29	0.00 ***
Conflicts with people	7.33	3.50	7.00	2.61	0.64	5.85	1.19	7.24	3.00	0.00 ***
Other disturbances										
Anxiety	9.05	3.88	7.52	3.37	0.08	6.60	1.77	8.49	3.20	0.00 ***
Mood swings	8.93	4.20	7.81	3.25	0.22	6.31	1.82	8.46	3.01	0.00 ***
Sensitive to rejection	7.53	3.69	6.84	3.18	0.41	5.98	1.69	7.40	2.75	0.00 ***
Less interest	8.92	4.34	7.84	3.06	0.23	6.97	1.88	8.87	3.21	0.00 ***
Difficulty concentrating	9.28	4.56	7.87	3.17	0.15	7.08	2.22	8.98	3.18	0.00 ***
Fatigue	11.95	6.11	11.32	3.93	0.62	8.03	3.13	12.54	3.95	0.00 ***
Increased appetite	9.43	5.40	9.16	3.93	0.82	8.35	3.26	11.03	4.66	0.00 ***
Crave specific foods	9.23	5.48	9.68	4.81	0.72	7.75	3.17	10.83	4.67	0.00 ***
Out of control	8.23	3.94	8.03	3.19	0.83	6.38	1.35	10.17	3.56	0.00 ***
Impairment in functioning caused by the symptoms										
Impaired work/daily routine	10.48	5.75	8.68	3.54	0.13	7.11	2.48	9.57	3.49	0.00 ***
Impaired hobbies/social	9.83	5.02	8.48	3.90	0.22	7.14	2.74	8.84	3.62	0.00 ***
Impaired relationships	8.23	3.73	8.19	3.60	0.97	6.08	1.98	8.46	2.89	0.00 ***

Note1. The *p*-values of differences between groups were calculated by independent two-sample *t*-test. Note2. The mean was the sum of DRSP scores of 5 days (the late corpus luteum) before menses. *** *p* < 0.001.

## Data Availability

Any data related to the study can be provided upon a reasonable request.

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
