# Peer review of "Effects of Yoga for Coping with Premenstrual Symptoms in Taiwan—A Cluster Randomized Study"

_healthcare, 2023, doi:10.3390/healthcare11081193_

Round 1
Reviewer 1 Report
Effects of yoga for coping with premenstrual symptoms in Taiwan
Women of reproductive age experienced premenstrual symptoms ranging up to 90%. Premenstrual symptoms has an impact on everyday life, regular activities, and interpersonal relationship. Stretching and breathing exercises, such as yoga and pilates could reduce the impact of the symptoms. Premenstrual symptoms are alleviated by yoga exercise directed by an experienced certified yoga instructor. The authors examine the effectiveness of home-based yoga practice on premenstrual symptoms, which is very interesting. This information could guide the debate and further research on this topic.
Title: The title is well chosen, reflecting the study being reported. The authors should considered adding a cluster randomized study to the title, reflecting the type of study they performed.
Overall:
The aim is well emphasized and explained. The paper is well written and was very pleasant to read.
Abstract:
No comments.
Introduction :
The introduction section is attractive to read, emphasizing the reason for conducting the study, explaining the background of premenstrual symptoms, also the relation to yoga exercises, which is excellent.
Material and Methods
This section is well written, explaining the method the study was conducting. Despite, three are several important and crucial points needing explanation and/or clarification.
1. In line 64 on page 2 the authors report the term convenient sample. This is not a familiar term . The term needs to be explained to the reader. Please do so.
2. In line 66-70 on page 2 the authors report the inclusion criteria of this randomized trial. The including women were not supposed to use any medication to reduce premenstrual symptoms, a particular group with premenstrual symptoms as they were not using any medication or management for their complaints. The question remains if this is a representative group to evaluate the effect of yoga treatment on premenstrual symptoms , please elucidate on this point.
3. The authors do not report if the women had previous treatment for premenstrual symptoms, which seems important for the interpretation of the results. Please elucidate thoroughly on this point.
4. The authors have chosen for three months follow-up. From a readers perspective three months seems a very short time frame to evaluate the result of the yoga treatment. Could the authors explain to the reader why three months were chosen as a time frame?
5. On page 2, line 73-78 results are presented, which should be reported in the result section. Please report the results in the result section.
6. The authors used the Daily Record of Severity of Problems (DRSP), I suppose this is a questionnaire?
7. The data were collected from March to December 2015, 8 years later. Could the authors explain why the results are published 8 years after collection?
Results
This section is well written, explaining the method the study was conducting. Despite, three are several important and crucial points needing explanation and/or clarification.
8. The authors should considered the use of a flow diagram to explain visualize the recruitment process. This helps the reader in the interpretation of the presented results.
9. Although the authors describe in the material and methods section that a question about whether yoga exercise was performed was added to the questionnaire to remind the participants to practice (line 103-104), the results are not presented . This again important information which should be reported to the reader. Please do so.
Discussion
This section is well written and very pleasant to read.
10. The authors report in line 166 that good compliance to the yoga exercises correspond with the effect. So, information regarding the yoga exercises compliance is very important in the interpretation of the presented results and for follow-up studies.
11. Other treatment modalities are not mentioned in this article. Are there comparisons to other treatment modalities? The authors should report this to the reader, as this is a relevant question.
12. One limitation is that the authors compares yoga exercises to placebo. Whether yoga exercises improves premenstrual symptoms compared to other treatment modalities is an important question which needs to be answered.
13. Furthermore, the short time frame should be considered a limitation. The effect of yoga exercises on the long term should be studied.
14. the specific group of women included, women not suppose to use any medication to reduce premenstrual symptoms should also be considered a limitation , as this is a very specific and particular group.
Reviewer 2 Report
Reviewer comments Manuscript # healthcare-2231156 “Effects of yoga for coping with premenstrual symptoms in Taiwan”
The premise of the paper- does regular self-directed yoga at home provide relief of PMS symptoms in women with PMS?- is valid particularly in light of the study cited in the introduction showing yoga classes with an instructor are beneficial
However, the clarity of the methods is seriously lacking and I believe the statistical analyses employed may be inappropriate for the data collected. Beyond this, the text could be significantly improved with editorial review by someone fluent in English
Major missing information:
Lin66-70 inclusion criteria do not specify enrolled women must have PMS symptoms. The authors need to demonstrate that the convenience sample obtained included women with PMS
Line 72- the power statement on study size is incomplete. It does not indicate how large a difference between groups was expected.
2.2 measures
Line 88-90 describe a 1to 6point Likert scale yet in table 2 most of the means presented in individual symptom line are larger than 6. Missing is a clear statement on how the data were treated.
Line 118-120 is not clear. Was ONLY data from the 5th day prior menstruation used? or were all days from 5days prior up to the day prior to menstruation summed and averaged?? Or was some other treatment of the data used?
Results
Baseline characteristics of the 2 groups is incomplete. No data are presented comparing baseline severity of PMS scores. As presented, we do not know if any of the women included in the study had PMS.
Further, without clarification that the two groups were not different at baseline with regards to symptom scores, one cannot conclude the intervention had an effect.
Table 1. is far less important than the space it takes up. Suggest authors use more descriptive language to establish the demographic characteristics of the groups were similar and use the space in the table to show DRSP baseline scores.
Table 2. I suspect the analyses used (independent T-test) is incorrect for the type of data collected. A Likert scale is usually evaluated treating the scale as ordinal- that is response categories are presented in a ranking order, but the distances between the categories cannot be presumed to be equal- and employ either Spearman’s or a chi-square test for independence. I strongly suggest the authors get a consult from a statistician as to the appropriate way to analyze their data. Given the number of analyses conducted and the likelihood many of the measured parameters are closely correlated more robust statistical tests may be needed.
If, as stated, the DRSP was collected daily by the women it would be useful to analyze the groups for DRSP scores on days of the cycle know NOT to be associated with PMS scores. This might also be an approach to establish that these women do in fact have PMS symptoms.
Minor missing information
Line 73-75 that exactly 128 women were listed as completing the study leaves me suspicious of exactly how enrollment was conducted. How many women were invited? How many were excluded (and for what reasons)? How many started but did not finish all 3 months? How many completed all 3 months?
Finally, the manuscript contains numerous errors in language & grammar. For example the sample described in line64 is a convenience sample not a convenient sample. Line 44-45 is missing the word “women” describing who benefits from yoga. Overall the reader gains a basic understanding of what the authors ree trying to communicate but a thorough editing by someone with native English language skills would dramatically improve the manuscript and increase its impact.

Round 2
Reviewer 2 Report
While the manuscript is somewhat improved, it remains insufficient in several areas.
Methods section 2.4 Data collection and Analysis
Lines 159-160 do not clearly state “the 5 days prior to menstruation were summed and averaged to provide a value for each variable in the DRSP.” Footnote in table 2 indicates this but this needs to be stated in this section of methods as well.
I do not see rationale for using the independent T-test for analysis. Based on my limited experience with Likert scales this may not be the appropriate statistical test. Spearman’s or a chi-square for independence may be more appropriate. As I stated in my first review:
A Likert scale is usually evaluated treating the scale as ordinal- that is response categories are presented in a ranking order, but the distances between the categories cannot be presumed to be equal- and employ either Spearman’s or a chi-square test for independence.
Results
The revised manuscript does not demonstrate that the two clusters had comparable baseline DRSP scores. Without this, there is simply no way to conclude the 20-25% improvements in DRSP for the Yoga group compared to non-yoga group is due to the intervention or merely a reflection of a in-going difference between the two groups. While table 1 indicates similar frequency of premenstrual pain, there is NO DATA presented showing DRSP scores at baseline.
Table 1 shows similar demographics for the two clusters but this does not mean they have similar PMS severity at the start of the study. I see no data establishing the two clusters are similar for PMS severity at baseline.
I might suggest using the cycle #1 DSP scores only to compare the two groups (Table 2 show all three cycles added and averaged) since it might be presumed any benefits of initiating regular exercise would have minimal effect after only a few days.
Alternatively, the authors could analyze DRSP scores on non-PMS days (for example compare the two clusters on cycle, 1 days 10-15). This is less direct but may provide some assurance the differences observed are not merely due to the convenience sample not having balanced PMS severity at study outset.
If the authors cannot show the two groups had similar PMS severity at study onset, then the paper can be considered only an exploratory pilot that suggests further carefully controlled study on the impact of yoga is warranted. This would require the conclusions of the manuscript be more speculative on the benefits of at-home yoga.
Finally, the English remains a major impediment to understanding the paper. For example
“Besides, women who met the criteria, and after understanding the aim of the 97 study were willing to participate in this study and complete a daily diary for three 98 menstrual cycles were recruited”
Would be better stated as
“Women who met the inclusion criteria were given an explanation of the study and needed be willing to complete a daily diary for 3 menstrual cycles.
Also, this statement
Considering avoiding women’s sharing each 99 other about the study intervention (yoga exercise) and premenstrual symptoms, 100 which might be kind of contamination of the study results. Hence, the two dis-101 tricts were clustered assigned to experimental and control group.
Could be improved as
In order to avoid women in the two treatment groups inadvertently sharing information on the intervention (yoga exercise), a cluster design employing two geographically distinct districts to recruit for the experimental and control groups was employed.
The manuscript has several places where the English can be dramatically improved and these are merely two examples.
A thorough review and edit by someone with a good grasp of English language and grammar would help.
